# Comparative Analysis of Chromatin-Delivered Biomarkers in the Monitoring of Sepsis and Septic Shock: A Pilot Study

**DOI:** 10.3390/ijms22189935

**Published:** 2021-09-14

**Authors:** Jesús Beltrán-García, Juan J. Manclús, Eva M. García-López, Nieves Carbonell, José Ferreres, María Rodríguez-Gimillo, Concepción Garcés, Federico V. Pallardó, José L. García-Giménez, Ángel Montoya, Carlos Romá-Mateo

**Affiliations:** 1Department of Physiology, Faculty of Medicine and Dentistry, University of Valencia, 46010 Valencia, Spain; jesus.beltran@ext.uv.es (J.B.-G.); E.Maria.Garcia@uv.es (E.M.G.-L.); Concepcion.Garces@uv.es (C.G.); Federico.V.Pallardo@uv.es (F.V.P.); J.Luis.Garcia@uv.es (J.L.G.-G.); 2INCLIVA Biomedical Research Institute, 46010 Valencia, Spain; edurnecarbonell@yahoo.es (N.C.); ferreresj@gmail.com (J.F.); mariarodriguezgimillo@gmail.com (M.R.-G.); 3Biomedical Research Networking Center of Rare Diseases (CIBERER), Institute of Health Carlos III, 28029 Madrid, Spain; 4Centro de Investigación e Innovación en Bioingeniería, Universitat Politècnica de València, 46022 Valencia, Spain; jmanclus@ci2b.upv.es; 5EpiDisease SL (Spin-Off Ciber-ISCIII), 46980 Paterna, Spain; 6Intensive Care Unit, Clinical University Hospital of Valencia, 46010 Valencia, Spain

**Keywords:** sepsis, septic shock, circulating histones, nucleosomes, HMGB1, biomarkers, ELISA

## Abstract

Sepsis management remains one of the most important challenges in modern clinical practice. Rapid progression from sepsis to septic shock is practically unpredictable, hence the critical need for sepsis biomarkers that can help clinicians in the management of patients to reduce the probability of a fatal outcome. Circulating nucleoproteins released during the inflammatory response to infection, including neutrophil extracellular traps, nucleosomes, and histones, and nuclear proteins like HMGB1, have been proposed as markers of disease progression since they are related to inflammation, oxidative stress, endothelial damage, and impairment of the coagulation response, among other pathological features. The aim of this work was to evaluate the actual potential for decision making/outcome prediction of the most commonly proposed chromatin-related biomarkers (i.e., nucleosomes, citrullinated H3, and HMGB1). To do this, we compared different ELISA measuring methods for quantifying plasma nucleoproteins in a cohort of critically ill patients diagnosed with sepsis or septic shock compared to nonseptic patients admitted to the intensive care unit (ICU), as well as to healthy subjects. Our results show that all studied biomarkers can be used to monitor sepsis progression, although they vary in their effectiveness to separate sepsis and septic shock patients. Our data suggest that HMGB1/citrullinated H3 determination in plasma is potentially the most promising clinical tool for the monitoring and stratification of septic patients.

## 1. Introduction

Management of sepsis remains a worldwide challenge due to the difficulty in early recognition of symptoms to facilitate stratification and accurate/proper treatment of septic patients [1]. For instance, progression from sepsis to septic shock is currently unpredictable and depends on predisposing factors of the patient, the severity of the infection, and the response to therapy, as well as the degree of organ dysfunction. Thus, clinicians face a complex diagnostic landscape in which early, quick, and efficient intervention remains a critical necessity.

In this regard, the detailed study of the complex molecular abnormalities underlying sepsis might provide interesting and promising molecular candidates to be used as sepsis progression biomarkers that can help define the process of septic staging, which is currently considered a clinical demand at intensive care units (ICUs). Inflammatory response-related molecules secreted in response to pathogens, together with the molecules and signaling cascades related to the immune system and coagulation factor activation, interact with the host’s cells to configure a complex network of molecules populating the blood of patients, which have been designated as pathogen-associated molecular patterns (PAMPs) and damage-associated molecular patterns (DAMPs). Among the latter, we find nuclear elements released to the bloodstream in different ways: first, as part of the process known as NETosis, the formation of neutrophil extracellular traps (NETs), wherein neutrophils release a mesh of altered chromatin and nuclear proteins during the innate immune response, which opsonizes and compromises pathogenic cell viability. Nuclear elements released from NETs affect neighboring cells, promoting endothelial cell necrosis and apoptosis and, hence, increasing the release of more nuclear content to the bloodstream via a second pathway which, in the end, leads to a positive feedback process that establishes a correlation between the increase in the release of nuclear content and disease progression. Given the tight relationship that exists between the production of free radicals and the generation of oxidative stress resulting from endothelial cell dysfunction, especially in the context of sepsis progression, focusing on the molecular features of nuclear elements released to the bloodstream is a source of substantial potential clinical interest [2]. Nuclear factors typically found in the bloodstream of sepsis patients include free DNA, mono-, di-, and oligo-nucleosomes, and free histone proteins, among other chromatin-derived elements; thus, these factors are acquiring increasing relevance as DAMPs. This has given rise to a rich field of research into the definition disease biomarkers [3,4,5,6,7,8,9]; however, the lack of performance when using any of these molecules on their own to discriminate between sepsis and other inflammation-related pathologies highlights the necessity of new and more specific biomarkers, which should also be preferentially used in combination [10]. The release of nuclear content has been documented as part of the inflammatory NETosis response and as a result of organ injury during sepsis progression [8,11]. In summary, histone proteins, either in form of NETs, nucleosomes, or as individual proteins, are responsible for endothelial cell damage by means of several transduction pathways which are not fully understood [12,13,14,15,16]. The cell death consequences of the presence of histones in blood suggest that they could be used as disease progression biomarkers [17,18]. In this regard, there is also evidence for the participation of another nuclear protein, High-Mobility Group Box-1 (HMGB1), as part of the cytokine storm triggered during the inflammatory response to infection [19]. HMGB1 contributes to the pro-inflammatory state, and several works have shown that extracellular HMGB1 levels correlate with disease progression and participate in cell death processes tightly linked to pyroptosis and endotoxemia [20]; however, many details regarding the specific kinetics of its release and clearance from the bloodstream are still partly unknown. Since HMGB1 can be found in differentially oxidized forms and has been reported to influence the activity of endothelial eNOS and ROS pathways [21], studying its presence in the bloodstreams of sepsis patients in relation to the impairment of clinical parameters could provide important clues regarding the regulation of antioxidant response homeostasis and sepsis progression.

Correlations between blood levels of nucleoproteins and disease severity [17,22,23] set the basis for detecting their circulating levels in the blood as a parameter that could predict the onset of sepsis in patients. Several detection methods to measure circulating histones, nucleosomes, and nucleoproteins have been developed in order to use these parameters as effective, time-saving biomarkers that could avoid the striking consequences of permanent histone levels in the blood or that could, at least, differentiate between those patients that should receive more aggressive treatments or earlier fluid resuscitation in order to avoid a fatal outcome. Most widely used methodologies involve immunoassays, generally in the form of enzyme-linked immunosorbent assay (herein, ELISA) kits. For example, one of the most used kits for the evaluation of circulating chromatin-derived molecules, Cell Death detection ELISAPLUS kit (Roche), consists of the detection of mono- and oligo-nucleosomes, but not cell-free circulating histones. Another similar parameter for which immunoassays have been developed is citrullinated histone H3, a specific post-translational modification that has been related to the correct and appropriate production of NETs during the innate immunity response as well as in the particular context of sepsis [24].

In the present work, we sought to evaluate the actual potential for decision making/outcome prediction of the most commonly proposed chromatin-related biomarkers (i.e., nucleosomes, citrullinated H3, and HMGB1) in a pilot study to differentiate between sepsis and septic shock patients in reference to controls, which included both healthy subjects and, importantly, critically ill nonseptic patients from the ICU who had suffered spontaneous intracerebral hemorrhage. To do so, we compare the results of quantification of these biomarkers obtained using different ELISA methods, including our own home-made nucleoprotein-detection ELISA kit, and analyzed the capacity of the different methods to discriminate between groups. Finally, we provide a series of correlation analyses between the aforementioned biomarkers and clinical parameters relevant to the monitoring sepsis and septic shock progression, emphasizing the use of specific chromatin-derived biomarkers for the particular subtypes of septic patients analyzed herein.

## 2. Results

### 2.1. Development of a Home-Made Immunoassay for Detection of Nucleoproteins in Plasma

In order to evaluate the potential of nucleoprotein detection in plasma as a sepsis biomarker, a home-made enzyme-linked immunosorbent assay (ELISA) based on monoclonal antibodies was developed. To this end, spleen cells from MRL-lpr/lpr mice that spontaneously produced antinucleoprotein antibodies were used to obtain monoclonal antibodies by applying the hybridoma technology. After fusions, clone selection was based on recognition of histone complexes as a protein core or as nucleosomes isolated from HeLa cells. Finally, 12 hybridomas were cloned and stabilized. Their corresponding monoclonal antibodies (MAb) were assayed for specificity towards HeLa nucleoproteins. Approximately half of the MAbs recognized both core histones and nucleosomes, while the other half were highly specific for nucleosomes. Next, MAbs displaying the highest titers for nucleoproteins were conjugated to peroxidase. After a thorough MAb examination in a sandwich ELISA format, a pair of MAbs showing high specificity to nucleosomes was selected for the immunoassay. The specificity of capture MAb L1–4 and detection MAb L2.14 is shown in Figure 1. As shown, the capture antibody recognized core histones, whereas the detection antibody was specific for nucleosomes. Next, immunoassay conditions such as immunoreagent concentrations, incubation times, and assay buffer were optimized. Finally, the optimized ELISA was evaluated in terms of sensitivity and reproducibility. A limit of detection (LOD) of 20 ng nucleosomes/ml was obtained (see Appendix B), with intra- and interassay coefficients of variation below 10%. As expected, the ELISA recognition pattern was that provided by the detection antibody (MAb L2–14), characterized by the specific recognition of nucleosomes and the lack of recognition of different combinations of histones, complex histone extracts, and DNA from different sources (Figure 1b). These results prompted us to test the kit using plasma samples from ICU patients, including patients with intracerebral hemorrhage (critically ill patients without sepsis), sepsis and septic shock patients, and healthy subjects, to validate the potential use of the tool for sepsis and septic shock progression monitoring.

### 2.2. Analysis of Nucleoprotein Levels in a Cohort of Sepsis and Septic Shock Patients

Fifty-three samples of blood from subjects of four different groups were analyzed using two different ELISA kits, as described in the Methods section. Clinical characteristics of the three groups of clinically ill subjects are described in Table 1. In general, the most significant clinical differences in Table 1 were found between control nonseptic patients and septic shock patients in all standard parameters. The measurements with statistically significant differences among groups were C-reactive protein (*p* > 0.0001), activated partial thromboplastin time (APTT) (*p* = 0.014), and the clinical APACHE II and SOFA scores (*p* > 0.03 and *p* < 0.001, respectively). It was noteworthy that mortality and ICU stay, and other classical biochemical biomarkers (i.e., lactate and PCT) did not show statistically significant differences between sepsis and septic shock patients. Interestingly, in the case of circulating nucleoproteins, the highest plasma levels were found in the septic shock group, followed by the sepsis group, when samples were analyzed utilizing both our home-made immunoassay (hereafter, Nucleosome kit 1) and the commercial Cell Death detection kit (Roche) (hereafter, Nucleosome kit 2) (Figure 2).

In order to compare the levels observed in the different groups, we performed nonparametric Kruskal–Wallis tests. For the Nucleosome kit 1, the differences between groups were statistically significant only between both control groups and septic shock patients (*p* < 0.01) (Figure 2a); the Nucleosome kit 2 differentiated healthy and septic shock subjects with the same statistical strength, but it reached greater statistical significance when separating healthy subjects and septic shock patients (*p* < 0.0001).

### 2.3. Citrullinated Histone H3 and HMGB1 Levels in Sepsis and Septic Shock Patients

Among the PAMPs and DAMPs associated with the challenging septic response, citrullinated histone H3 and HMGB1 have been suggested as potential sepsis biomarkers. Hence, we decided to use a second set of commercial kits for the determination of these molecules in the same samples from the four cohorts. As found with the other kits used in this work, the highest citrullinated H3 levels were observed for septic shock cases (Figure 3a). Differences were statistically significant between healthy subjects and ICU controls and septic shock (*p* < 0.0001), and between healthy subjects and ICU controls and sepsis (*p* < 0.05). Regarding determination of HMGB1 levels, and in contrast to the previous results, these were not significantly higher exclusively in septic shock patients; in fact, significance between healthy subjects and sepsis was higher than between healthy subjects and septic shock patients (*p* < 0.0001 vs. *p* < 0.001); when comparing ICU controls and both groups of patients, again the highest statistical significance was found between ICU controls and sepsis patients (*p* < 0.001). In contrast, ICU controls and septic shock patients were differentiated with *p* < 0.01 (Figure 3b). It should be noted that although all previously analyzed molecules exhibited the highest levels in the group of septic shock patients, in the case of HMGB1, the levels were comparable in sepsis and septic shock patients, and it was the only case in which the mean value was higher in the sepsis group as compared to the septic shock group (Figure 3c).

### 2.4. Comparative Analysis of the Diagnostic Potential of Chromatin-Delivered Biomarkers

In order to further compare the different ELISA methods, a ROC curve analysis was performed to evaluate the diagnostic power of the levels of nuclear proteins and citrullinated histone H3 as relevant biomarkers to distinguish septic shock or sepsis cases from critically ill ICU patients suffering not from infectious processes, but from spontaneous intracerebral hemorrhage (Figure 4). The AUCs (areas under ROC curves), standard error, confidence interval (CI), optimal concentration cut-off value, and sensitivity and specificity percentages for each kit were calculated to differentiate between controls and cases (sepsis and septic shock patients together) (see table in Figure 4b). Interestingly, levels of HMGB1 stood out with a 100.0% sensitivity and 90.0% specificity to differentiate between control and case groups, indicating that the presence of HMGB1 is a reliable biomarker for the septic state as compared to critically ill noninfected individuals from ICU, and thus supporting its use as an early marker of sepsis. Citrullinated histone H3 also provided a good sensitivity (81.5%) and as much specificity (100.0%) as the nucleosome-based detection methods, although the latter failed in terms of sensitivity, which was below 80% for both nucleosome kits. It should be pointed out that the highest statistical confidence was found for citrullinated histone H3 and HMGB1 (*p* < 0.0001).

### 2.5. Correlation with Clinical Parameters in Sepsis Patients Varies between Nucleosomes, Citrullinated Histone H3, and HMGB1

Extracellular histones have been previously related to sepsis progression and organ failure, specifically linked to proinflammatory and prothrombotic effects [18,25,26,27]. It has also been suggested that HMGB1 could be used as a late marker of sepsis [28], although our results showed increased levels of HMGB1 not only in septic shock but in sepsis patients. In order to extract as much information as possible about the differential behaviors of chromatin-delivered biomarkers during sepsis progression, we calculated the correlation coefficients between each of the biomarkers in our study and clinically relevant parameters of the studied cohorts (see Appendix A for a complete correlation matrix). The higher number of correlations was found among the group of sepsis patients (Figure 5). Interestingly, the pattern of positive and negative correlations was relatively similar for both nucleosome-specific kits and the CitH3 kit, and slightly different from the pattern observed for HMGB1. When analyzing the specific significance of these correlations, differences were highlighted among all kits, providing specific positive and negative correlations: we found very strong correlations between CitH3, total SOFA score, and MAP; and between HMGB1 and other important clinical parameters like ICU LOS and DD (Figure 5a). These results, taken together with the previous analysis of diagnostic capacity and mean values found in plasma from the different groups of patients, highlight that CitH3 and HMGB1 levels stand out as the most informative biomarkers in terms of stage of the sepsis process and relevant clinical parameters related to sepsis management, with CitH3 being a potential better biomarker for the diagnostic of septic shock and HMGB1 a more prognostic one; in fact, we generated ROC curves to evaluate the predictive potential of HMGB1 levels for ICU stays longer than 5 days, obtaining an AUC value of 0.9583 and a specificity and sensitivity of 83.33% and 100%, respectively, for levels higher than 15.27 ng/ in sepsis, but no significant cut-off value was obtained for its use as a reliable predictor in the group of septic shock patients (Figure 5b). Finally, it was noteworthy that the only biomarker significantly increased in the specific subset of nonsurviving patients was citrullinated H3 (14.11 ng/ in surviving patients versus 19.00 ng/ in nonsurvivors; *p* = 0.03); this allowed us to create a ROC curve using citrullinated H3 level as a criterion to differentiate surviving sepsis and septic shock patients from those nonsurviving, obtaining an AUC of 0.897 with an standard error of 0.062, a CI95% of 0.7745 to 1.000, and sensitivity and specificity values of 83.3% and 81.0%, respectively, using a cut-off value of 14.81 ng/(*p* value = 0.003) (Figure 5c).

## 3. Discussion

Sepsis management remains a worldwide critical healthcare problem, despite striking advances in diagnostic and pharmacological interventions that have lowered the incidence during the last twenty years [29]. Nonetheless, deaths by sepsis are usually underestimated given the high burden of the disease in low-income countries, where documentation of cases remains uncertain or incomplete. In this particular context, the capacity to distinguish the cryptic first signs of septic processes from other conditions and, significantly, to predict if a septic process is on the verge of becoming a septic shock, constitutes a hotspot in the clinical management of the critically ill patient. The presence of extracellular nucleoproteins, either as free histones or in the form of nucleosomes, has been shown to correlate with disease severity and to participate actively in the pathogenicity, being critical determinants of organ failure and hence indicators of bad prognosis [25,30]. The use of histone-based biomarkers, however, has never been standardized and incorporated into clinical routine, although novel methods based either in immunological detection [15,25] or in mass-spectrometry detection, as we have previously published [22], are currently available [31].

In this work, we evaluated the potential of several markers based on different chromatin-delivered analytes to differentiate sepsis and septic shock patients, compared to healthy individuals and critically ill nonseptic control patients from ICU. One of the immunoassays most commonly found in the literature is the Cell Death Detection ELISAplus, previously used to assess nucleosome levels in critically ill patients [23,30]. However, this kit does not provide a standard curve to obtain objective quantification of plasma circulating nucleoproteins, unlike our home-made ELISA kit with high specificity towards human nucleosomes. Nonetheless, neither of both tests showed the capacity to differentiate between sepsis and septic shock patients. Similarly, we detected significantly higher CitH3 levels in septic shock patients as compared to both control groups; this is of especial relevance since a previous work by Li et al. identified high blood levels of citrullinated H3 after lipopolysaccharide injection in a rodent model to produce septic shock [24]. Interestingly, when we measured the protein HMGB1 in human plasma, we found the maximum increase in the group of sepsis patients. Although no statistical significance was found between sepsis and septic shock patients, mean values were higher in the sepsis group. HMGB1, released to the bloodstream during the initial inflammatory cascade, has been postulated as a pro-inflammatory cytokine that could serve as an early biomarker in sepsis. However, its differential functions regarding oxidation status, cellular localization, and release to the bloodstream make it a difficult target for therapeutic intervention [20,28]. To assess the feasibility of the methods herein assayed for use with diagnostic or prognostic objectives, we performed ROC curve analysis to obtain specificity and sensitivity values for all ELISA kits. The test which showed the best performance to differentiate all control individuals from sepsis and septic shock patients was the analysis of HMGB1, with a specificity of 81.8% and a specificity of 96.0%, in agreement with the highest levels found in both groups of patients as compared to the rest of biomarkers assayed. These results are interesting given that the groups of clinically ill patients did not differ strikingly in the levels of classical sepsis biomarkers like lactate or PCT, and the fact that mortality rates were also quite similar. However, diagnostic and/or prognostic potential of biomarkers relies strongly upon the correlation between their levels and the different clinical parameters that allow monitoring of disease progression and severity; in our study, we found a strong correlation between all chromatin-delivered biomarkers and relevant clinical parameters, but especially important are those high and significant correlations found between CitH3, total SOFA score, and MAP, and between HMGB1, ICU LOS, and DD. Previous works support the relationship between HMGB1 levels and cardiovascular impairment [32,33,34]. It is interesting to point out that the levels detected in our results were significantly higher in septic patients even when using the group of ICU patients suffering from spontaneous intracerebral hemorrhage as a control group. It is noteworthy that HMGB1 has been shown to be present in differentially oxidized and reduced forms [21,28], but little information has been provided regarding the inflammatory properties of the different forms when found in the extracellular milieu and in the context of sepsis progression, where oxidation conditions are significantly increased; thus, further investigation will be required to ascertain the usefulness of the analysis of specific forms of HMGB1 as well as the therapeutic potential of antioxidant therapies in mitigating the deleterious effects associated with increased plasma levels of HMGB1. Although the correlation between HMGB1 levels and sepsis progression is not novel in itself, as shown by Karlsson and collaborators [35], there have been very few works, to our knowledge, that compare the levels of this biomarker among patients classified according to the latest SEPSIS-3 diagnostic criteria (i.e., sepsis vs. septic shock) and, importantly, compared to both healthy individuals and noninfected ICU patients. In our work, plasma samples were collected from patients at 24 h after ICU admittance, in contrast to the cohort analyzed in [35] which were collected at 48 h post-admission, probably explaining the lack of correlation of HMGB1 with the outcome in their results. Finally, our work compared in parallel the levels of three different types of chromatin-related biomarkers.

Taken together, these results seem to suggest that the kinetics of release and presence in the bloodstream for HMGB1 and histone-related molecules (i.e., nucleosomes and citrullinated H3) are different, and point to a landscape in which monitoring of HMGB1 levels is more useful for earlier stages of sepsis diagnosis and to predict long ICU stays, whereas increased levels of CitH3 would be more reliable for monitoring and prediction of septic shock onset, organ failure, and death. It should be noted that, given the limitations of our work (reduced sample number), the actual values of circulating histones and nucleoproteins should be further calculated with more quantitative techniques; in this regard, mass-spectrometry-based technologies constitute a promising path to follow, as we previously proposed [22]. However, the present results point to a relevant usefulness of the measurement of CitH3 and HMGB1 by ELISA methods, which are more accessible to low-income facilities, which could also benefit from home-made immunoassays addressed to detect nucleoproteins, as we have shown.

In conclusion, these results suggest that all biomarkers analyzed show a very similar distribution pattern, with HMGB1 showing the highest levels in sepsis patients as compared to septic shock ones. All in all, the presence of nucleosomes and specific modified forms of histone proteins together with other chromatin-derived nuclear proteins involved in inflammation rise as relevant markers of disease severity, and refinement of their detection and quantification methods would be of capital relevance towards their future incorporation into the clinical routine.

## 4. Materials and Methods

### 4.1. Production of Monoclonal Antinucleoprotein Antibodies

MRL-lpr/lpr female mice (9 weeks old) were from Harlan Laboratories. These mice spontaneously develop an autoimmune disease characterized by circulating antinuclear antibodies and immune complex glomerulonephritis (see Appendix B for details). Animal manipulation was carried out following the Spanish regulations currently in force and under the approval of the Ethical Committee for Research of Universitat Politècnica de València, and all methods were carried out according to their corresponding guidelines and regulations.

### 4.2. Home-Made Nucleosome ELISA

The immunoassay consisted of a sandwich format using MAb L1–4 as the capture antibody and MAb L2–14 as the detection antibody. For this purpose, MAb L2–14 was previously conjugated to peroxidase using the HRP Conjugation Kit (Abcam, Cambridge, UK), following the manufacturer’s instructions. For the immunoassay, a volume of 100 µL per well was used for all assay steps. After each incubation time, plates were washed four times with washing buffer (PBS containing 0.05 Tween 20). First, ELISA plates (Costar #3590, Corning, NY, USA) were coated overnight at room temperature with MAb L1–4 at 2 µg/ in 50 mM carbonate buffer (pH 9.6). Next, plasma samples diluted 1:5 in assay buffer and controls were added and plates were incubated for 1 h. After washing, HRP-MAb L2–14 at 0.5 µg/ in assay buffer was added and incubated for 1 h. Finally, peroxidase activity was determined by adding the substrate solution (2 mg/o-phenylenediamine and 0.012% H_2_O_2_ in 25 mM citrate-62 mM phosphate, pH 5.3). After 10 min, the reaction was stopped with 2.5 M sulfuric acid and the absorbance at 490 nm was read and recorded with a SpectraMax 190 microplate reader (Molecular Devices, San José, CA, USA). Proteins and combination of proteins plus DNA used to test the specificity of the ELISA are described in Appendix B.

### 4.3. Commercial ELISA Kits

The commercial ELISA kits used in this work were as follows: Cell Death Detection ELISAPLUS kit (Roche, Basel, Switzerland) for photometric determination of mono- and oligonucleosomes by combining an anti-histone-biotin-antibody and an anti-DNA-peroxidase antibody; EpiQuickTM Circulating Histone H3 Citrullination ELISA kit (EpiGentek, Farmingdale, NY, USA), in which histone proteins contained in plasma samples are captured on strips coated with anticitrullinated histone H3 antibody, detected by colorimetry and quantified thanks to an internal standard curve; and HMGB1 ELISA (IBL International, Hamburg, Germany), with strips coated with anti-HMGB1 purified antibody, detected by colorimetry and quantified thanks to an internal standard curve. Assays were performed following manufacturers’ instructions, using 20 µL of plasma (1/4 dilution) for the Cell Death Detection kit, 30 µL of plasma for the citrullinated H3 detection kit, and 70 µL of plasma for the HMGB1 detection kit. Read-outs of the plates was obtained by measuring absorbance at 405 nm and 450 nm, respectively, using a SpectraMax 190 microplate reader (Molecular Devices, San José, CA, USA).

### 4.4. Selection of the Cohorts and Blood Collection

To provide an accurate evaluation and comparison for the usefulness of the three studied ELISA kits in the classification of sepsis and septic shock patients, we selected a cohort of individuals (*n* = 53) divided into four different groups. All of the subjects participating in this project signed the informed consent form. The first group was of healthy subjects (*n* = 17), a group in which no significant presence of nucleoproteins in plasma should be expected, composed of 10 male and 7 female volunteers (age: 49.76 ± 5.18 years); next, a cohort of control nonseptic patients (*n* = 9) from the ICU of the Clinical University Hospital of Valencia (HCUV) who were not affected by infection, autoimmune disease, or polytrauma injury but by spontaneous intracerebral hemorrhage, were set as a control group in which a certain level of nucleoproteins, free DNA, and histones could be present due to organ injury. Finally, two different cohorts including sepsis (*n* = 10) and septic shock patients (*n* = 17) with confirmed bacteremia (microbiological blood positive culture at 48 h) were included, all from the same ICU as the previously mentioned control group (see Appendix B for details on exclusion criteria).

Peripheral blood samples were collected using EDTA tubes from both healthy controls and ICU patients. In the latter, blood samples were collected within the first 6 h after being admitted to ICU. Each sample was centrifuged at 2500 rpm for 10 min at room temperature (RT) to separate plasma. Aliquots were then stored at −80 °C until analysis.

### 4.5. Statistical Analysis

Descriptive analysis was performed by calculating parameters such as mean, median, standard deviation, and confidence intervals. Kruskal–Wallis and post hoc tests were used for comparisons between groups. ROC analyses were performed by calculating AUCs (areas under ROC curves), standard error, confidence interval (CI), optimal concentration cut-off value, and sensitivity and specificity percentages for each kit with diagnostic and stratification purposes. *p* values of < 0.05 were regarded as being statistically significant. All the analyses were conducted using SPSS, v.24 (IBM Corporation, Armonk, NY, USA).

## Figures and Tables

**Figure 1 ijms-22-09935-f001:**
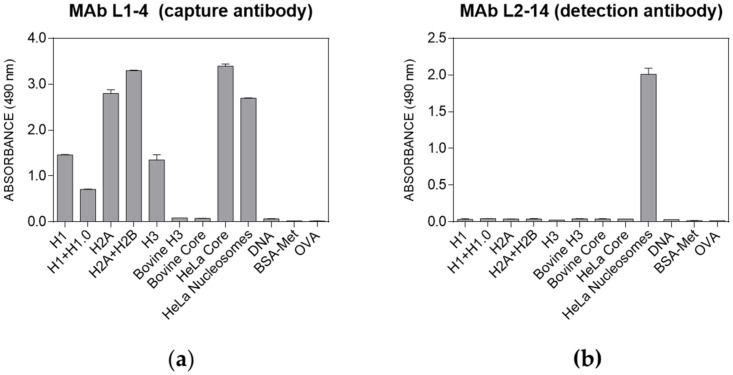
Specificity patterns of the monoclonal antibodies selected for the nucleoprotein sandwich ELISA. (**a**) The MAb L1–4 capture antibody recognized all histone fractions from human origin. (**b**) In contrast, the MAb L2–14 detection antibody showed a reaction only to HeLa nucleosomes (BPS Bioscience). Data represent mean values +/− SD (*n* = 2). H1, H1.0, H2A, H2B, H3, and HeLA core correspond to histones from human origin; DNA to calf thymus DNA; BSA-Met to albumin methylated from bovine serum; and OVA to albumin from chicken egg. See Appendix B for more details on the proteins assayed.

**Figure 2 ijms-22-09935-f002:**
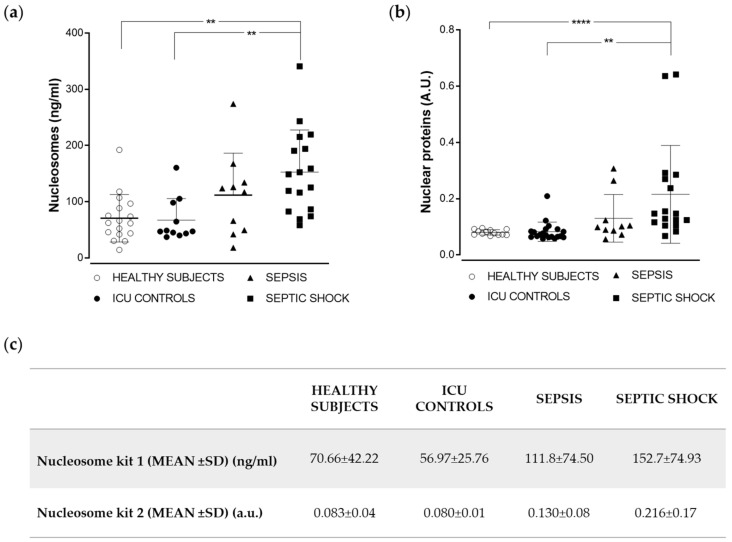
Measurement of nucleoproteins using ELISA kits. Plasma samples were analyzed using a home-made ELISA kit, Nucleosome kit 1 (**a**) and a commercial ELISA kit (Cell Death Detection ELISAPLUS kit; see Methods), Nucleosome kit 2 (**b**) specific for mono- and oligo-nucleosomes. Horizontal lines represent average values for each group, and bars represent standard deviation. Numerical values are summarized in (**c**). (**** *p* < 0.0001; ** *p* > 0.01).

**Figure 3 ijms-22-09935-f003:**
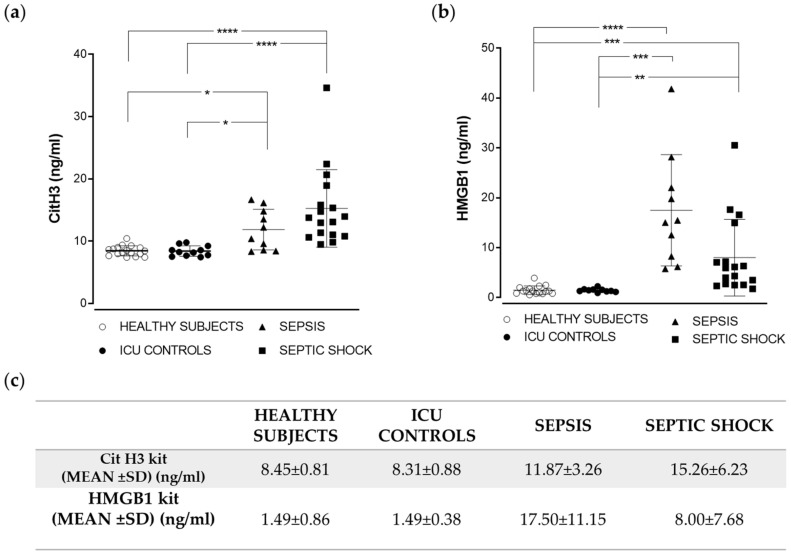
Measurement of citrullinated histone H3 levels (CitH3) (**a**) and HMGB1 (**b**) in plasma samples using commercial ELISA kits (see Materials and Methods for details). Horizontal lines represent the average values for each group, and bars represent standard deviation. Numerical values for both parameters are summarized in (**c**) (**** *p* > 0.0001; *** *p* < 0.001; ** *p* > 0.01; * *p* < 0.005).

**Figure 4 ijms-22-09935-f004:**
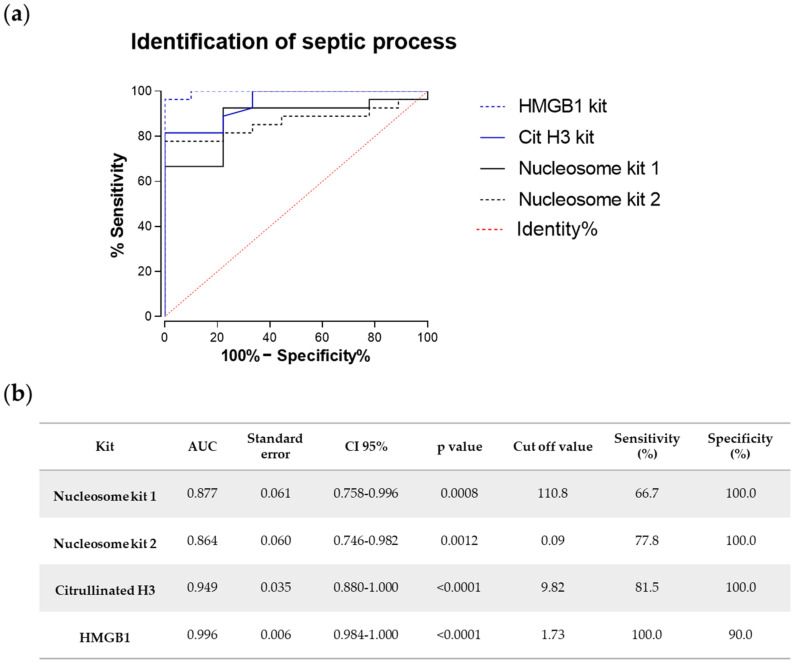
Diagnostic capacity of ELISA methods to differentiate control and septic cases. (**a**) ROC curves for nucleoproteins, HMGB1, and citrullinated H3 levels, measured using the three different ELISA approaches, as biomarkers for diagnosis of septic processes. ICU controls were compared to septic patients (sepsis patients plus septic shock patients). (**b**) Parameters and statistical significance obtained for the different ROC curves.

**Figure 5 ijms-22-09935-f005:**
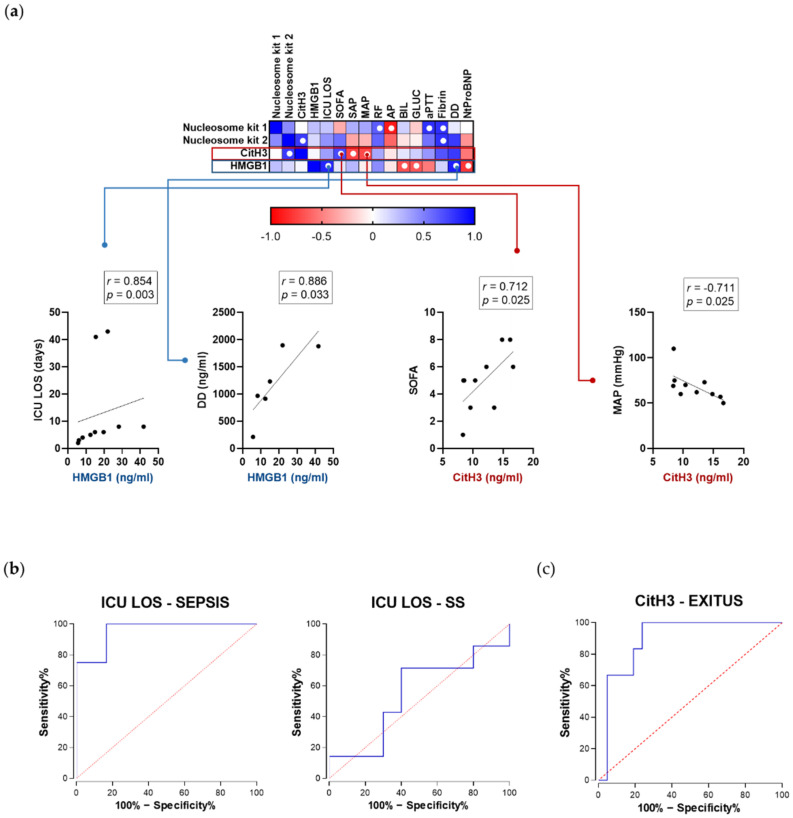
(**a**) Correlation between biomarker levels measured using the different ELISA kits and clinical parameters. Nonparametric Spearman correlation coefficients were calculated among the different variables: blue cells represent positive correlations and red cells represent negative correlations (see graphic legend); white dots indicate r values higher than 0.6, which reached the minimum significance (*p* ≤ 0.05). The top correlations most significant for CitH3 and HMGB1 are shown in detail. ICU LOS: ICU length-of-stay; SOFA: total SOFA score; SAP: systolic arterial pressure; MAP: mean arterial pressure; RF: respiratory frequency; AP: alkaline phosphatase; BIL: total bilirubin; GLUC: glucose; aPTT: activated partial thromboplastin time; Fibrin: fibrinogen; DD: D-dimer; NtProBNP: amino-terminal fragment of the pro-natriuretic peptide type B. All clinical parameters represent values at day 1. See Appendix A for detail on the complete r and *p* values for each correlation. (**b**) ROC curves obtained using HMGB1 levels to predict ICU stays longer than 5 days in sepsis (**left**) and septic shock patients (**right**). (**c**) ROC curve obtained using CitH3 levels to predict fatal outcomes for sepsis and septic shock patients.

**Table 1 ijms-22-09935-t001:** Baseline clinical features of the septic patients and ICU controls at admission ^1^.

	Control Nonseptic ICU (*n* = 9)	Septic ICU (*n* = 10)	Septic Shock ICU (*n* = 17)	*p* Value
**Demographics and clinical indexes**				
**Age (years) (mean ± SD)**	64.38 ± 8.31	65.10 ± 13.10	66.18 ± 11.60	ns
**Male gender (%)**	6 (66.7)	7 (70.8)	10 (58.8)	ns
**APACHE II score (mean ± SD)**	15.11 ± 4.80	15.80 ± 4.90	22.12 ± 9.07	0.03
**SOFA (mean ± SD)**	3.44 ± 3.09	5.00 ± 2.21	9.65 ± 3.43	<0.0001
**Organ support therapy (1st day)**				
**Vasopressor therapy (%)**	1 (11.1)	2 (20.0)	16 (94.1)	<0.0001
**CRRT (%)**	0	0	2 (11.76)	ns
**Mechanical ventilation (%)**	3 (33.3)	0	3 (21.4)	ns
**Inflammatory parameters**				
**White blood cells (mean ± SD)**	10,603 ± 4939	12,977 ± 11,589	16,395 ± 9612	ns
**CRP (mg/L) (mean ± SD)**	25.14 ± 26.31	250.20 ± 133.46	339.35 ± 189.72	<0.0001
**PCT (ng/mL) (mean ± SD**	0.06 ± 0.07	2.21 ± 3.08	1.52 ± 2.29	ns
**Lactate**				
**Lactate 1st hour (mmol/L) (mean ± SD)**	1.61 ± 6.98	1.91 ± 1.25	1.81 ± 1.99	ns
**Coagulopathy parameters**				
**Platelets count/L (mean ± SD)**	240.4 × 10^3^ ± 89.3 × 10^3^	229.6 × 10^3^ ± 147.36 × 10^3^	169.4 × 10^3^ ± 122.8 × 10^3^	n.s
**APTT (seconds) (mean ± SD)**	29.67 ± 2.88	99.833 ± 118.82	117.43 ± 154.83	0.014
**Outcome**				
**ICU LOS (days) (mean ± SD)**	5.9 ± 6.6	12.6 ± 15.62	9.47 ± 8.61	ns
**Hospital LOS (days) (mean ± SD)**	15.4 ± 14.6	20.6 ± 13.1	19.88 ± 17.8	ns
**ICU Mortality (%)**	2 (22.2)	2 (20.0)	4 (23.5)	ns

^1^ ICU: intensive care unit. APACHEII: Acute Physiology and Chronic Health Evaluation score during the 1st day of ICU admittance. SOFA score: Sequential Organ Failure Assessment score during the 1st day of ICU admittance. CRRT: continuous renal replacement therapy. CRP: C-reactive protein. PCT: procalcitonin. APTT: activated partial thromboplastin time. LOS: length of stay. Significant differences (*p* value < 0.05) between groups.

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
