# Peer review of "Comparative Analysis of Chromatin-Delivered Biomarkers in the Monitoring of Sepsis and Septic Shock: A Pilot Study"

_ijms, 2021, doi:10.3390/ijms22189935_

Round 1

Reviewer 1 Report

In this study the authors aimed to compare different Elisa methods for chromatin-derived biomarkers in order to discriminate sepsis from non-sepsis patients.

Unfortunately some issues make this manuscript not suitable for publication in this journal.

Firstly, there is no novelty. One of the main findings of the present study is that HMGB1 is the most reliable biomarker of those studied, and that it increases in sepsis patients. It has already been shown that HMGB1 concentrations are elevated in 247 sepsis patients (although no association with the outcome was shown, PMID: 18297269).

The number of patients included in the study is really small. Moreover, in Methods the described cohort includes 61 individuals, but in Results line 143 we read that 53 samples were analysed.

Did these patients receive corticosteroids? Was this a limiting factor for inclusion in the study?

Author Response

In this study the authors aimed to compare different Elisa methods for chromatin-derived biomarkers in order to discriminate sepsis from non-sepsis patients.

Unfortunately some issues make this manuscript not suitable for publication in this journal.

Firstly, there is no novelty. One of the main findings of the present study is that HMGB1 is the most reliable biomarker of those studied, and that it increases in sepsis patients. It has already been shown that HMGB1 concentrations are elevated in 247 sepsis patients (although no association with the outcome was shown, PMID: 18297269).

We thank Reviewer 1 for their comments and suggestions. Regarding the novelty of our work, we would like to highlight that, although we agree with the reviewer that the correlation between HMGB1 levels and sepsis progression is not novel on itself, there are very few works, to our knowledge, that compare the levels of this biomarker among patients classified according to the latest SEPSIS-3 diagnostic criteria (i.e. sepsis vs septic shock); and importantly, comparing also to both healthy individuals and ICU non-infected patients (the controls in the report quoted by the reviewer (PMID: 18297269 ) are healthy individuals, not ICU controls). Another difference is that in our work, plasma samples were collected from patients at 24h since ICU admittance, in contrast to the cohort in PMID: 18297269 which were collected at 48h post-admission, probably explaining the lack of correlation of HMGB1 with the outcome, which we were able to found. Finally, our work compares in parallel the levels of three different types of chromatin-related biomarkers.

Our results show that the levels of HMGB1 exhibit a markedly different pattern as those of Citrullinated histone H3, which shows higher levels in septic shock patients as compared to sepsis ones. To further corroborate this observation, we have re-elaborated Figure 4 in our manuscript, removing the group of healthy individuals in order to check if the biomarkers analyzed are still potentially good to be used as specific biomarkers for cases of infection, as compared to non-infected ICU controls which could also show increased levels of inflammatory markers. The data still corroborate that both HMGB1 and Citrullinated H3 are the best biomarkers for septic processes as compared to circulating nucleosomes. Furthermore, the strong correlation found for clinical parameters like DD levels and, especially, the strong correlation between HMGB1 and ICU LOS, in contrast with the other specific correlations found for Cit H3 (including the capacity to distinguish the group of non-suriving patients), are also in our opinion a useful addition to the current knowledge of the diagnostic utility of each of these biomarkers. We would like to point out that from our work the following relevant result can be obtained: HMGB1 levels are more efficient to be used in the monitoring of sepsis patients rather than to septic shock patients, which is also a novel addition to the field; although we admit the limitation of our small sample size, we think the significant differences found between the different biomarkers are quite interesting. Moreover, the relationship between Citrullinated histone H3 levels and not only multiorgan failure but fatal outcome (see new lines 300-307 and new Figure 5c) is also a relevant result that opens new paths for investigation, suggesting better prognostic potential that the analysis of circulating nucleosomes. Since all these biomarkers, as measured by ELISA methods, are potentially transferable to the clinical routine, this can be of interest for clinical use and management of sepsis and septic shock patients.

Finally, we would like to point out that the presentation of the ELISA kit developed by our group, which uses a standard curve in order to provide absolute nucleosome values, means an improvement over the more extended commercial kit and provides a useful method for researchers in order to develop their own nucleosome detection methods, which is also an addition to the current solutions commercially available.

The number of patients included in the study is really small. Moreover, in Methods the described cohort includes 61 individuals, but in Results line 143 we read that 53 samples were analysed.

Following the well-founded worries of the reviewer about the number of patients enrolled in the study, we have changed the title accordingly and we have highlighted that our report is a pilot study. So, the new title is: “Comparative analysis of chromatin-delivered biomarkers in the monitoring of sepsis and septic shock. A pilot study”. Nonetheless, we would like to highlight that the specific value of our work is the comparison between closely related-yet not fully investigated biomarkers and the potential to easily detect their levels by ELISA methods, either commercial or home-made.

We also thank Reviewer 1 for their observation and apologize for the mistake in the number of patients in the cohort; we have corrected the manuscript to make sure the numbers are consistent.

 Did these patients receive corticosteroids? Was this a limiting factor for inclusion in the study?

We thank reviewer 1 for underscoring this very important issue. This was not a limiting factor for inclusion in the study, and corticosteroids were used when needed according to clinical criteria. Nonetheless, no statistically significant differences were found related to the biomarkers studied among patients treated or not with corticosteroids. We think that, in agreement with the reviewer’s observation, a larger cohort study including comparative monitoring of diverse treatments from the perspective of the levels of these biomarkers would be of great relevance, especially focusing in citrullinated H3 and HMGB1 as our results suggest they are more likely to correlate with organ failure and fatal outcome.

Reviewer 2 Report

The authors describe using a combination of analytical methods including a home-made ELISA-assay the usefullness of determination of plasma-circulating histons for digosing procedure of sepsis and prediction of clinical course. The design is interesting, but it is not new. There are some considerations for interpretation of the results:

  • from a quite clear pathomechanistical point of view you delineate your question, but the primary aim of the study needs some clarification: to detect a cut-off level for nucleosomes for clinical decision making/monitoring of treatment - its confusing.
  • there are no data given with respect to evaluation of the homemade ELISA-test (precision, intraday variance, lower limit of detection etc.) comparing to the commercially available kit.
  • paired comparison of data with the two tests might improved the quality of presentation (Fig. 2). Please use also an absolute quantity for 2b.
  • the study cohort is rather small, there is no difference between the lactate levels between the septic patients and the patients undergoing septicc shock.  This is surprising and should be discussed.
  • grouping of helathy controls and ICU-controls (without infection doesn't make any sense (Fig. 4). The strategy should be designed to distinguish patients with sepsis at ICU, not from HC. 
  • are there any data describing the clinical course and effectiveness of treatment wrt to histone levels? 
  • Please prove and discuss whether a combination of classifiers might be beneficial to aswer the underlying question.
  • for verification of results a subset of trauma patients with and without sepsis should be included.

Presentation of your interesting data needs some improvement, thus in consequence major revision prior to publication is required. 

Author Response

The authors describe using a combination of analytical methods including a home-made ELISA-assay the usefullness of determination of plasma-circulating histons for digosing procedure of sepsis and prediction of clinical course. The design is interesting, but it is not new. There are some considerations for interpretation of the results:

We thank Reviewer 2 for considering that our study is interesting; we would like to point out that, even though the biomarkers themselves are not novel, they have never (at least to our knowledge) been analyzed in parallel in septic and non-septic ICU controls, as we present in our work.  Thus, we think it provides new insights in several key points when studying sepsis and septic shock: as already mentioned, using two types of controls (healthy volunteers and ICU controls) is very important, since most of the studies performed in sepsis and septic shock patients are compared with healthy volunteers. The capability to stablish a good chromatin-derived biomarker for sepsis must be challenged with other ICU patients who could also present increased levels of nucleoproteins due to traumatic injury, as we perform in our work. In addition, although several papers analyze the presence and progression of HMGB1 levels in serum/plasma, we herein show that HMGB1 is able to be associated with the ICU  stay of sepsis patients and, importantly, that its levels are highly increased already in sepsis patients; thus, from the comparison with other chromatin-derived markers analyzed in the same sets of patients we provide evidence that HMGB1 and Citrullinated H3 are potentially better biomarkers to predict ICU length of stay and multiorgan failure/exitus, respectively (see correlations found for Cit H3 and SOFA scores).

from a quite clear pathomechanistical point of view you delineate your question, but the primary aim of the study needs some clarification: to detect a cut-off level for nucleosomes for clinical decision making/monitoring of treatment - its confusing.

We thank the reviewer for highlighting this inconsistency in our work: the primary aim of the study was to evaluate the actual potential for decision making/outcome prediction of the available chromatin-related candidate biomarkers (i.e. nucleosomes, citrullinated H3 and HMGB1) by ELISA measuring. Some of them have already been tested on their own, but to our knowledge their levels have rarely been compared in the same subsets of patients, and neither their correlation with clinical parameters usually measured in critically-ill patients admitted in the ICU. We thus think that the novelty and strength of our work is this comparative feature, which is further increased by the fact that we compare the most commonly used ELISA detection method for nucleosomes (the commercial Cell Death kit from Roche) with a home-made assay that proves to be similarly efficient and, moreover, provides absolute values since we use a standard curve. However, in both cases none of these measurements equals the capacity of HMGB1 and Citrullinated H3 to predict specific outcomes and disease progression, although these biomarkers have rarely been proposed for use in the clinical routine for septic processes.

In order to clarify the goal and purpose of our work, we have re-written part of the summary (see lines 24-26 and 31-32) as well as the introduction (see lines 108-110).

there are no data given with respect to evaluation of the homemade ELISA-test (precision, intraday variance, lower limit of detection etc.) comparing to the commercially available kit.

We appreciate the reviewer’s suggestions. As regards to the requested comparison of our home-made ELISA for nucleosome detection with the commercially available kit, both methods are not directly comparable in terms of limit of detection (LOD), because in the Cell Death kit this parameter is referred as number of apoptotic cells/mL, whereas our ELISA uses a standardized nucleosome preparation from HeLa cells and the LOD is therefore expressed in ng of nucleosomes/mL. Both methods are not either comparable in terms of precision and/or variability, since data for the commercial one are not included in the product specifications. As for our ELISA method, it showed intra- and inter-assay coefficients of variation below 10%, which are standard for ELISA techniques in clinical diagnosis.

Nevertheless, in order to improve the clarity of the paper, section 2.1 devoted to our home-made ELISA has been partially rewritten, now including new ELISA specification data. Besides, the HeLa nucleosome ELISA standard curve has been incorporated in Appendix A (see re-written section A.3) because we consider this information not as relevant to be included in the “core” text of the manuscript.

paired comparison of data with the two tests might improved the quality of presentation (Fig. 2). Please use also an absolute quantity for 2b.

We totally agree with the reviewer’s observation that a paired comparison of both methods would be adequate; nonetheless, as we have explained in the previous answer this is not possible since the commercially available kit (Cell Death by Roche) already harbors the limitation that it fails to provide absolute nucleosome levels; actually, the method is said to detect nuclear proteins and no standard curve is ever provided to define whether it is more specific for mono- or oligonucleosomes, or nuclear histone, as we have tested with our method (see Figure 1 and new figure in Appendix A.3). Besides, the dynamic ranges of both kits are different and as such we think that comparison of results using arbitrary units might not be a good option to reach our goal, which is testing the potential for these kits to be used in the clinical routine with confidence, regardless of the units in which the levels are expressed (although obviously availability of actual absolute levels are always an advantage). Under this comparison, although both kits provide similar specificity and sensitivity values (Figure 4) ours provides quantitative data which we consider more adequate, especially in order to compare to other chromatin-delivered nucleoproteins like citrullinated H3 and HMGB1.

the study cohort is rather small, there is no difference between the lactate levels between the septic patients and the patients undergoing septicc shock.  This is surprising and should be discussed.

We thank the reviewer for their comment: we have modified the title to highlight that this is a pilot study, since to our knowledge no other work compares four different methods to detect different nucleoproteins in plasma from four different group of healthy and critically ill individuals. We also want to underscore that although the number of patients is low this is a rather homogenous cohort if you compare it with other reports; the reviewer is right that the lactate levels are quite similar between the two groups of sepsis patients. Nonetheless, other criteria (as based in SEPSIS-3) for septic shock diagnosis are reached, since all of them required vasopressor therapy to maintain MAP over 65 mmHg. We think that our results corroborate the difficulty to classify sepsis patients and to distinguish or prognosticate progression from sepsis towards septic shock; in this context, analyzing the levels of nucleoproteins and their correlation with clinical parameters in such a homogenous set of septic patients provides insightful directions for clinical management that could be used to guide further and larger-cohort studies, starting from our preliminary pilot study.

grouping of helathy controls and ICU-controls (without infection doesn't make any sense (Fig. 4). The strategy should be designed to distinguish patients with sepsis at ICU, not from HC.

We thank the reviewer for addressing this point. The reviewer is right and we have re-elaborated Figure 4 in the new version of the manuscript to compare septic patients to ICU controls. The new data corroborate the best performance of Citrullinated H3 and HMGB1 to differentiate septic patients, as compared to nucleoproteins by both ELISA methods analyzed; we have also re-written the text in the corresponding results section (see lines 207-218).

are there any data describing the clinical course and effectiveness of treatment wrt to histone levels? Please prove and discuss whether a combination of classifiers might be beneficial to aswer the underlying question.

The reviewer is totally right that the analysis of correlations between treatment and biomarker levels would be very useful; however, we did not found statistically significant correlations with treatments received by these set of patients, although following the advice of the reviewer we compared the different nucleoprotein levels in the specific subgroup of non-surviving patients and found that citrullinated levels of histone H3 were higher (14.11 ng/mL versus 19.00 ng/mL) with a statistical significance of p = 0.03). Besides, we also performed ROC curves with the levels of the different biomarkers analyzed comparing the group of surviving and non-surviving patients, and interestingly we found a significant cut-off only for citrullinated H3 levels to predict fatal outcome. We have included this information in section 2.5 in the new version of the manuscript (lines 300-308 and new Figure5c).

Regarding the combined use of biomarkers, following the suggestion of the reviewer we attempted to create logistic regression models combining the different biomarkers analyzed, but they failed to provide statistical significance. Nonetheless, given that our work is a pilot study, we think the results that we present (which show that citrullinated H3 and HMGB1 levels are potentially the best biomarkers to follow the progression of septic patients) set the path for a larger study which could be performed using only these two biomarkers in order to evaluate outcomes and monitoring treatments, and maybe then, a more robust combined predictive model could be designed.

for verification of results a subset of trauma patients with and without sepsis should be included.

We thank the reviewer for their interesting suggestion. In the ICU control group of our study we have included patients who were not affected by infection, autoimmune disease or polytrauma injury (see clarifications added in section 4.4), but from spontaneous intracerebral hemorrhage; we excluded polytraumatic patients because there is literature supporting evidence of increased levels of histones in these patients that could confound the comparative analysis that we ought to perform ( see PMID: 32403440 or PMID: 23220920). The inclusion of a subgroup of patients presenting both traumatic injury and sepsis, after having tested the efficiency of citrullinated H3 and HMGB1 to monitor sepsis and septic shock progression, would be a really interesting project to perform but goes beyond the scope of the present manuscript.

Presentation of your interesting data needs some improvement, thus in consequence major revision prior to publication is required.

We really thank the reviewer for their insightful considerations, that clearly have improved the quality of our report. We hope that all the changes introduced in the second version of the manuscript could reach the high standards of IJMS.

Reviewer 3 Report

Interesting work regarding the use of nucleosomes, citrullinated H2 and HMGB1 in the setting of patients admitted to the ICU for sepsis monitoring.

Data has been presented at one time point for patients that are admitted to the ICU (various disease states) and compared to healthy subjects.

  • It has been noted that these patients appear to have an established diagnosis of either Non-infectious Critical Illness, Non-Shock-Sepsis (severe Sepsis) and Septic Shock.
  • This should be stated more clearly.

Table 1 contains baseline clinical features which are very helpful for clinicians to relate to these patients.

  • Could you provide information regarding the source of the infection and culture positivity in a supplement. The table you provide is sufficient, but it may be helpful to have this additional information in the supplement.

Line 281 Figure 5 (a) has some formatting issues specifically in the top heat map graph with overlaying text. CitH3 also has several formatting issues.

  • Please reformat and possibly enlarge for better visibility.

Line 361. The word “exitus” should be replaced by the world death.

Author Response

Interesting work regarding the use of nucleosomes, citrullinated H2 and HMGB1 in the setting of patients admitted to the ICU for sepsis monitoring. Data has been presented at one time point for patients that are admitted to the ICU (various disease states) and compared to healthy subjects.

It has been noted that these patients appear to have an established diagnosis of either Non-infectious Critical Illness, Non-Shock-Sepsis (severe Sepsis) and Septic Shock.

This should be stated more clearly.

 We thank the reviewer for their kind comments and positive evaluation of our work.

Regarding the classification of our cohort, patients have been diagnosed based in the latest diagnostic criteria established after SEPSIS-3 consensus, into sepsis and septic shock patients. Non-infectious critical patients, all of which suffered from spontaneous intracerebral hemorrhage, are described in section 4.4 and were considered as “ICU controls”; nonetheless, and in order to clarify from the beginning of our exposition the nature of this group of patients, we have also added this information in lines 112-113, 145, and  242-243 of the manuscript.

Table 1 contains baseline clinical features which are very helpful for clinicians to relate to these patients.

Could you provide information regarding the source of the infection and culture positivity in a supplement. The table you provide is sufficient, but it may be helpful to have this additional information in the supplement.

We agree with the reviewer that this is a relevant information, which we have added into an expanded section A.4 in the Appendix (please see lines 575, and 584-587). 

Line 281 Figure 5 (a) has some formatting issues specifically in the top heat map graph with overlaying text. CitH3 also has several formatting issues. Please reformat and possibly enlarge for better visibility.

 We thank the reviewer for their appreciation; we have reformatted and expanded the figure and we hope it is clearer in its current form. Nonetheless, a high resolution version of the figure will also be provided to the journal editors.

Line 361. The word “exitus” should be replaced by the world death.

We thank the reviewer for noticing this inconsistency in the manuscript; we have changed the word as suggested, as well as in figure 5c.

Round 2

Reviewer 2 Report

Thank you for considering all of my suggestions, from my point of view I have no further comments.

Author Response

Thank you so much for the acceptance of our corrections, and again, for making such an improvement on our work with your comments and suggestions.